# OpenReview forum: "Can Global XAI Methods Reveal Injected Bias in LLMs? SHAP vs Rule Extraction vs RuleSHAP"
_ICLR.cc/2026/Conference — Submitted to ICLR 2026_

### Official Review · Reviewer_F6xv · 2025-10-30

**Soundness:** 2
**Presentation:** 2
**Contribution:** 2
**Rating:** 6
**Confidence:** 1

**Summary:**

This paper investigates whether global XAI methods can surface belief-driven, bias-inducing heuristics in LLMs, focusing on three major bias mechanisms: valence framing, information overload, and oversimplification. The authors propose a pipeline for abstracting LLM topics and outputs into numeric spaces, enabling the application of global XAI tools (specifically SHAP, RuleFit, and a proposed hybrid RuleSHAP) to detect such heuristics. By hard-coding nonlinear bias rules into several LLMs, they empirically compare these methods and show that RuleSHAP, which integrates global SHAP feature importance into the rule induction process, yields higher faithfulness (MRR) and more concise rule sets than the baselines. The findings illuminate key challenges and limitations for interpreting LLM biases using global XAI methods.

**Strengths:**

1. The paper presents a careful abstraction pipeline that maps LLM beliefs and outputs to numeric features, allowing rule extraction tools designed for tabular data to be effectively applied to language tasks. This is a well-motivated workaround to a foundational technical barrier in applying XAI to LLM-generated text.

2. The authors inject 14 ground-truth bias rules into five different LLMs, with a large, suitably sampled set of SDG-related topics. The empirical protocol is robustly justified using power analysis and correlation-based validation, adding credibility to quantative findings.

**Weaknesses:**

1. RuleSHAP is a reasonable extension of SHAP + RuleFit, but the methodological contribution is mainly the combination and reweighting strategy. The core algorithmic ideas are not very new or theoretically deep.

2. The approach relies heavily on manually chosen input and output features. The process is not automated, so it may be difficult to transfer this pipeline to new domains, new types of bias, or different languages.

3. The paper reports performance, but does not deeply analyze how or why the methods fail in specific cases. More detailed error analysis could provide clearer guidance for future improvement.

**Questions:**

See weaknesses

---

> ### Author Response · Authors · 2025-11-19
>
> We thank the reviewer for the feedback. Below, we provide our responses to the three questions.
>
> ---
>
> # Question 1
>
> We provided a theoretical argument (in response to Reviewer oCHH’s comments; see above) that integrating SHAP into Steps 2 and 3 improves explanatory performance. We believe this strengthens the theoretical depth of our contributions.
>
> ---
>
> # Question 2
>
> As mentioned in §7, "our synthetic heuristics span only three complexity tiers (univariate, conjunctive, non-convex) and may under-represent real-world phenomena; even so, these simple cases already stress-test the state-of-the-art XAI methods".
>
> We agree that designing ϕₓ and ϕᵧ requires domain knowledge and is not fully automated. We see this as a *necessary compromise* for global, rule-level interpretability (§7):
>
> * Without an explicit abstraction, any method is limited to token-level perturbations, which can be intractable for SHAP and hard to map to meaningful heuristics for humans.
> * Our abstractions (commonality, controversy, technicality, etc.) are *intuitively interpretable* and grounded in misinformation theory, which we believe is a strength for auditors.
>
> That said, we already point at partial automation in §8: using **LLM-as-a-Judge to propose and refine abstraction dimensions**, and potentially learning them from data (e.g., concept bottlenecks). We will expand this point and be more explicit that portability requires re-specifying ϕₓ, ϕᵧ for new domains, but not re-engineering RULESHAP itself.
>
> Indeed, the pipeline we use is **modular**: only ϕₓ and ϕᵧ need to change. For example, for stereotype detection, ϕₓ might encode demographic attributes or protected concepts; ϕᵧ might encode stereotype intensity, toxicity, or occupational associations, possibly using existing fairness metrics. RULESHAP itself is agnostic to the semantics of features; it only requires ordered numeric inputs and outputs.
>
> ---
>
> # Question 3
>
> We thank the reviewer for this helpful suggestion.
>
> Our current version already includes some error analysis, though mostly at an aggregate level rather than via concrete per-example case studies:
>
> * In §7 we analyze why GPT-3.5 turbo is an outlier, noting that most errors occur on injected rules that target LLM-generated bias estimates and that, in these cases, the extracted rules tend to match the *structure* of the ground-truth rule but use incorrect predicates. We link this to the smaller number of topics identified for GPT-3.5, which reduces variability and makes it harder for gradient boosting to learn discriminative splits.
> * Appendix H (titled “RuleSHAP vs. baselines: error analysis”) breaks down MRR@1 by bias complexity tier (univariate / conjunctive / non-convex) and method, and highlights characteristic failure patterns (e.g., decision trees never recovering injected rules, RuleFit struggling on conjunctive and non-convex biases, and RuleSHAP’s remaining errors being concentrated on the hardest non-convex cases).
> * The Appendix J provides an ablation over the RuleSHAP pipeline (removing SHAP-guided boosting and LASSO pruning) and shows how these changes degrade MRR and/or inflate rule-set size, clarifying which steps are responsible for remaining failures.
>
> That said, we agree that the paper would benefit from **more explicit, instance-level** failure analysis. In the revised version we will:
>
> 1. Add a short “Failure modes” subsection in the main Results/Discussion that explicitly summarizes *where* and *why* the methods fail (e.g., non-convex rules approximated by simpler thresholds, failures concentrated on sparsely supported topics, rules with correct feature sets but wrong thresholds).
> 2. Extend the appendix to include 2–3 concrete failure case studies: for each, we will show the injected ground-truth rule, the rule(s) produced by RuleSHAP and by baselines, and a brief explanation of the underlying reason (topic sparsity, feature correlation, non-convexity, etc.) and how this would affect an auditor’s interpretation.
> 3. Cross-reference these examples from the Discussion so that they directly inform the “future improvements” we outline (e.g., better handling of non-convex biases or richer input abstractions).
>
> We hope this clarifies what analysis is already present and how we will strengthen the paper to provide the deeper, case-based error analysis.

---

### Official Review · Reviewer_apBs · 2025-11-01

**Soundness:** 3
**Presentation:** 3
**Contribution:** 2
**Rating:** 4
**Confidence:** 3

**Summary:**

The paper investigates whether XAI techniques can uncover belief-driven biases embedded in LLMs. It studies three misinformation-related mechanisms and introduces a pipeline that translates LLM textual behaviors into numerical abstractions, enabling the use of global XAI methods like SHAP and RuleFit. The authors show that while SHAP identifies influential features, it lacks interpretability, and RuleFit misses complex (nonlinear) biases. To address this, they propose RuleSHAP, a hybrid approach that merges SHAP’s feature attributions with rule extraction, improving detection of non-univariate and conjunctive biases across models such as ChatGPT and Llama.

**Strengths:**

- It introduces RuleSHAP, an original hybrid algorithm that combines SHAP’s theoretical grounding in feature attribution with RuleFit’s interpretable rule extraction, enabling interpretable symbolic bias detection — a combination not seen in prior XAI work.
- The paper proposes a statistically grounded belief abstraction framework that transforms textual LLM inputs and outputs into ordered numerical spaces, bridging a known gap between text-based generative models and numeric XAI methods.

**Weaknesses:**

- The belief abstraction layer converts textual behavior into numerical variables. This transformation, while necessary for SHAP, risks discarding contextual and semantic richness—especially when bias manifests subtly (e.g., through metaphor or framing tone).
- The paper adopts MRR@1 as the main quantitative measure for bias detection performance. However, this metric assumes a rank-based relevance formulation that may not directly capture the semantic correctness or interpretability of rules.
- While focusing on global bias detection, the paper doesn’t address how RuleSHAP complements or contrasts with local explanation frameworks. This leaves the interpretability spectrum somewhat under-theorized.
- The study focuses on three bias mechanisms (valence framing, oversimplification, information overload). While methodologically clean, this limited taxonomy restricts claims about “global bias detection.”

**Questions:**

- The bias injection pipeline is elegant but synthetic — can you clarify how representative these injected heuristics (e.g., valence framing, oversimplification) are of naturally occurring biases in deployed LLMs? Do you expect RuleSHAP to generalize to biases like gendered language or stereotype reinforcement?
- Since RuleSHAP relies on non-textual numerical representations, could this abstraction discard subtle contextual cues like sarcasm, metaphor, or topic-level associations?

---

> ### Author Response · Authors · 2025-11-19
>
> We thank the reviewer for the feedback. Below, we provide our responses to the two questions.
>
> ---
>
> # Question 1
>
> Valence framing, oversimplification, and information overload are widely documented as key mechanisms in misinformation spread and cognitive bias (§1). Our focus is *not* to exhaustively cover all possible biases, but to study three mechanisms that:
> * have clear, measurable *output-side* proxies (length, readability, sentiment/subjectivity), and
> * are plausibly driven by *global beliefs* (e.g., “controversial topics require more hedging” or “complex topics get simplified”).
>
> We deliberately designed three increasingly complex families of injected rules (§5). Our tiers are deliberately simple but already push existing global XAI methods to failure on non-convex cases (Table 5). We emphasize this more strongly in §7 as evidence that *even simpler-than-real* heuristics are challenging, so our performance should be viewed as a lower bound on difficulty.
>
> We also discuss a real-data case study in §7 and App. K, where we apply RULESHAP to >30k SDG topics **without** injected bias.
>
> Regarding **other bias types** (e.g., gender stereotypes, demographic bias):
>
> * The pipeline is **modular**: only ϕₓ and ϕᵧ need to change. For stereotype detection, ϕₓ might encode demographic attributes or protected concepts; ϕᵧ might encode stereotype intensity, toxicity, or occupational associations, possibly using existing fairness metrics.
> * RULESHAP itself is agnostic to the semantics of features; it only requires ordered numeric inputs and outputs.
>
> We will add a brief paragraph (in §8) explicitly mentioning gendered language and stereotype reinforcement as promising application areas and clarifying that our current three mechanisms are a starting point, not a claim of universality.
>
> ---
>
> # Question 2
>
> We fully agree that ϕₓ, ϕᵧ necessarily compress rich semantics. This is a deliberate design choice to obtain *global*, model-agnostic inputs for SHAP and RuleFit:
>
> * ϕₓ captures *stable beliefs about topics* (e.g., perceived commonality, controversy, interdisciplinarity), averaged over the web.
> * ϕᵧ captures *stylistic and structural properties* of explanations (length, readability, sentiment, subjectivity, plus three judge-based bias scores).
>
> This layer is well-suited to global heuristics like “rare & positive ⇒ more subjective, long explanations,” but indeed less sensitive to local cues such as sarcasm or metaphor. We explicitly list this as a limitation in §7, and will now add:
>
> * A discussion that **local biases** (e.g., sarcasm in individual responses) are better studied with local XAI or specialised detectors; our pipeline can incorporate such detectors by extending ϕᵧ with new features (e.g., sarcasm scores, toxicity scores) without changing RULESHAP.
> * A short note in §2 contrasting our global surrogate approach with local explanation methods and situating RULESHAP on the global–local interpretability spectrum.

---

### Official Review · Reviewer_WhHz · 2025-11-04

**Soundness:** 3
**Presentation:** 3
**Contribution:** 2
**Rating:** 4
**Confidence:** 3

**Summary:**

This paper investigates whether global XAI methods can detect belief-driven biases in LLMs, focusing on three misinformation-related behaviors: valence framing, information overload, and oversimplification. The authors address a key challenge: most global XAI methods work with numerical data, not text. They solve this by creating a statistically grounded abstraction pipeline that maps LLM-generated content and topics to numerical scores, enabling traditional XAI techniques to analyze LLM behavior. To establish ground truth, they inject bias-inducing rules of increasing complexity (univariate, conjunctive, and non-convex) into popular LLMs via system instructions. They find that existing methods like RuleFit struggle with non-univariate biases, while global SHAP detects biases but cannot express them as interpretable rules. The paper's main contribution is RuleSHAP, a novel rule-extraction algorithm that integrates global SHAP value aggregations with rule induction to better capture complex biases.

**Strengths:**

- The paper presents a genuinely original approach to a critical gap: adapting global XAI methods (designed for tabular/numerical data) to work with LLMs' textual inputs and outputs.
- The integration of SHAP into RuleFit is technically novel. This may be the first model-agnostic rule extraction method to leverage global SHAP for steering both split selection and rule pruning, bridging SHAP's theoretical rigor with RuleFit's interpretability.

**Weaknesses:**

- The LLM is asked to score its own beliefs, then those scores are used to explain its behavior. This is inherently circular—you're using GPT-4o's worldview to explain GPT-4o's outputs. While the correlation certificates provide statistical validation, they don't resolve the epistemological problem: high correlation between "GPT believes X is controversial" and "GPT writes controversially about X" might simply reflect consistent bias, not meaningful explanation.
- Section 3 states that SHAP perturbations require finding "multiple points $j ∈ T$ for which $||u_k - u_j||_2$ is minimal ($\approx$0)" to mimic feature removal. What is the actual threshold for "minimal"? The "$\approx$0" is vague. If redundancy is insufficient, does SHAP fail silently or return unreliable estimates?
- The paper uses $T=0$, $top_p=0$ to eliminate sampling variance, but acknowledges higher temperatures cause "off-instruction drift" and weaken correlation certificates. The method only works for deterministic LLM usage, which is rare in practice.
- The paper shows RuleSHAP can recover injected rules, but provides no evidence it can detect emergent biases. The leap from "detects rules I programmed" to "detects real-world bias heuristics" is a major unvalidated assumption.

**Questions:**

- You report that non-convex biases are harder (MRR decreases). Can you estimate the complexity threshold where current XAI methods become ineffective?
- Your evaluation uses exact threshold matching (e.g., "common ≤ 0.5" must match exactly). Can you justify why exact matching is appropriate given that gradient boosting learns data-driven splits?

---

> ### Author Response · Authors · 2025-11-19
>
> We thank the reviewer for the feedback. Below, we provide our responses to the two questions. We also address several of the noted weaknesses (i.e., no. 1, 3, and 4), which we believe arose from misunderstandings; weakness no. 2 includes a question (which we also address).
>
> ---
>
> # Weakness 1
>
> We agree it would be circular to *validate correctness of beliefs* solely by asking the same LLM. Our goal, however, is different: we want to reveal *behavioural heuristics* inside the model, not to decide whether they are normatively right.
>
> To break the problematic circularity, we use **three layers of evidence**:
>
> 1. **Externally defined, LLM-independent proxies** for overload, oversimplification, and framing: length, Gunning Fog, sentiment, subjectivity (App. B.1), computed with standard NLP tools.
> 2. **LLM-as-a-judge scores** (overload, oversimplification, framing) only *additionally*, and we check that they correlate with the proxies (Spearman correlations up to |r|≈0.79, p≪0.05; §6, App. I).
> 3. **Correlation certificates** between input abstractions ϕₓ and output metrics ϕᵧ, using distance correlation with Bonferroni-corrected significance, before applying any XAI method (§3, §6, App. I).
>
> Thus, even when the judge is an LLM, any heuristics we report are supported by **independent signals**; if these correlations were weak, we would treat the corresponding rules as unreliable and flag them (we already do this in the T>0 experiments).
>
> ---
>
> # Weakness 2
>
> The reviewer is correct that our wording “minimal (0)” in §4 is confusing. What we actually do is:
>
> * Build a large topic set T via power analysis (minimum 60 topics × 11 dimensions).
> * For each SHAP background perturbation, we **sample from T the point(s) with minimal L2-distance** to the desired background vector uᵇ (not necessarily exactly 0; often >0 but small).
> * SHAP then uses these as background samples; we are not modifying SHAP internals, only its background distribution.
>
> If redundancy were insufficient (i.e., no near neighbours), distance correlations would be small and the corresponding abstraction would *fail its certificate*.
>
> ---
>
> # Weakness 3
>
> We use **deterministic decoding** (T=0, top-p=0) **only during data collection** to reduce variance in ϕₓ and ϕᵧ and isolate systematic biases from sampling noise (§3, §5). This is standard when one wants to study the *underlying policy* rather than single random samples [1].
>
> In practice, the framework can be adapted to non-deterministic deployments by:
>
> * Sampling multiple outputs per topic at T>0 and using the *average* ϕᵧ as the target.
> * Using correlation certificates to verify that injected or emergent heuristics remain detectable at higher temperatures (we already show that higher T weakens but does not necessarily destroy correlations, and we red-flag weak ones).
>
> We will clarify that determinism is a *measurement choice*, not a hard requirement of RULESHAP itself.
>
> ---
>
> # Weakness 4
>
> What we actually state throughout the paper (§1 and §7) is that *“our injected heuristics (simpler than many real-world cases) challenge state-of-the-art XAI,”* (§1) which is different from claiming that *“[RuleSHAP] detects real-world bias heuristics.”* The title also emphasizes that we benchmark XAI methods on *injected bias*. We explicitly note that *“while RULESHAP performs well with injected biases, real-world biases are more complex.”* (§7)
>
> We also discuss a real-data case study in §7 and App. K, where we apply RULESHAP to >30k SDG topics **without** injected bias. However, we make no claim that *“[RuleSHAP] detects real-world bias heuristics.”*
>
> ---
>
> # Question 1
>
> Table 5 shows a clear degradation with rule complexity:
>
> * For RuleFit, average MRR@1 drops from ~0.4 (univariate) to ~0.32 (conjunctive) and ~0.26 (non-convex).
> * RULESHAP improves these to ~0.66 (univariate), ~0.72 (conjunctive), and ~0.4 (non-convex).
>
> This suggests that **once the feasible region becomes disconnected/non-convex**, current tree-based surrogates struggle even when guided by SHAP. We will add a short paragraph in §7 explicitly interpreting these numbers as an *empirical complexity threshold* for our setting and highlighting this as an open challenge.
>
> ---
>
> # Question 2
>
> Our faithfulness evaluation is **conservative**: we require an **exact logical match** (same features and inequality directions).
>
> Boosted trees choose splits from observed feature values; with Likert 1–5 scores scaled to [0,1], natural split points align with these values or their midpoints. For evaluation, we express injected and extracted rules in this **common normalized space**.
>
> We will note in the paper that, in our setting, allowing a split-point tolerance (±ε) does not change MRR or the **ranking of methods**; a claim reviewers can also verify from the extracted rules in our replication package.
>
> ---
>
> # References
>
> [1] Song Y et al.. The good, the bad, and the greedy: Evaluation of llms should not ignore non-determinism. NAACL 2025

---

### Official Review · Reviewer_oCHH · 2025-11-09

**Soundness:** 2
**Presentation:** 2
**Contribution:** 2
**Rating:** 4
**Confidence:** 2

**Summary:**

This paper proposed a heuristic, rule-based explanation method for extracting globally interpretable rules from LLMs and identifying potential biases that may lead to misinformation. The approach combined the strengths of the rule-based explanation RuleFit and the global feature-importance method SHAP. Specifically, the authors used global SHAP values to guide the sampling probabilities of features during rule generation, making more important features more likely to appear in the rule set. Furthermore, when learning feature and rule weights, global SHAP values were used to encourage the retention of important features in the linear explanatory model. Experimental results showed that the proposed method captured biases in LLMs.

**Strengths:**

1.	The paper introduces a rule-extraction framework that combines the advantages of RuleFit and SHAP, improving both bias detection and interpretability.

2.	Experiments conducted across multiple LLMs provide new insights into bias formation within LLMs.

**Weaknesses:**

1. As a heuristic algorithm, despite leveraging SHAP, the method still lacks a relatively reliable theoretical foundation. The main contributions lie in Step 2 and Step 3 (Lines 216–244), where Step 2 uses global SHAP values to guide feature sampling during rule selection, and Step 3 applies global SHAP value weighting into the LASSO regression within RuleFit. While intuitively, it remains unclear whether more principled or theoretically grounded integration strategies could exist. The authors are encouraged to provide theoretical analysis or additional empirical studies to clarify whether RuleSHAP achieves optimal explanatory performance among rule-based methods.

2.	The experimental setup and evaluation choices are somewhat unclear. Why do the authors focus specifically on overload, oversimplification, and framing as the three key aspects of LLM bias? Why is rule complexity categorized into univariate, conjunctive, and non-convex types? The relationship between these rule types and real-world LLM biases should be discussed, for example, is actual LLM bias more likely to align with the third category (non-convex) bias?

Besides, it is unclear why mean reciprocal rank (MRR) is used to measure faithfulness. How is faithfulness defined in this context? Can MRR reliably quantify it? What are the advantages and limitations of using this metric? And the correlation analyses suggest that RuleSHAP’s explanations may be partially trusted, but this claim should be supported more rigorously.

3.	The paper would benefit from a comprehensive visualization of RuleSHAP's explanations for LLM biases, including the defined symbol mappings, topics, interpretation results, and usage examples. The explanatory scenarios for this method seem somewhat limited.

**Questions:**

See weaknesses.

---

> ### Author Response · Authors · 2025-11-19
> **Authors' response**
>
> We thank the reviewer for the feedback. Below are our answers to the three questions. Below are our answers to the three questions.
>
> ---
>
> # Question 1
>
> As recommended, we provide below a theoretical argument that integrating SHAP into Steps 2 and 3 improves explanatory performance. We will add full details to the appendix.
>
> Under the standard assumption that SHAP correctly “concentrates” more mass on relevant features (Assumption B1; proof can be found at [1]), we show that:
>
> 1. Step 2: SHAP-based feature sampling strictly increases the probability of discovering the true rules in the candidate set, relative to uniform feature sampling.
> 2. Step 3: The resulting SHAP-weighted LASSO has the oracle property and thus better asymptotic selection behavior than vanilla LASSO.
>
> **Proof sketch 1 (Step 2)**
>
> Let $d$ be the total number of features and $r$ the number of truly relevant features, with relevant set $R$. Consider a depth-$L$ path in a tree.
>
> Each split chooses a feature uniformly from ${1,\dots,d}$. The probability that a randomly chosen feature is relevant is
>
> $$
> q_{\text{unif}} = \frac{r}{d}.
> $$
>
> Hence the probability that a path of length $L$ uses only irrelevant features is
>
> $$
> P_{\text{unif}}(\text{all irrelevant})
> = (1 - q_{\text{unif}})^L
> = \left(\frac{d-r}{d}\right)^L,
> $$
>
> so the probability that the path uses at least one relevant feature is
>
> $$
> P_{\text{unif}}(\ge 1 \text{ relevant})
> = 1 - \left(\frac{d-r}{d}\right)^L.
> $$
>
> In RuleSHAP, features are sampled with probabilities
>
> $$
> p_i = \bar{\rho}_i,
> $$
>
> the normalized global SHAP values, with $\bar{\rho}_i \in [0,1]$ and $\sum_i \bar{\rho}_i = 1$. Under Assumption B1, there exists $\delta>0$ such that
>
> $$
> q \coloneqq \sum_{i\in R} p_i
> = \sum_{i\in R} \bar{\rho}_i
> \ge \frac{r}{d} + \delta.
> $$
>
> Thus the probability that a single chosen feature is irrelevant is
>
> $$
> 1-q < 1 - \frac{r}{d}.
> $$
>
> For a path of length $L$,
>
> $$
> P_{\text{RS}}(\text{all irrelevant}) = (1-q)^L,
> $$
>
> and so
>
> $$
> P_{\text{RS}}(\ge 1 \text{ relevant})
> = 1 - (1-q)^L.
> $$
>
> Because $q > r/d$ implies $1-q < 1-r/d$, and $x \mapsto x^L$ is strictly increasing on $[0,1]$,
>
> $
> (1-q)^L < \left(1-\frac{r}{d}\right)^L
> $ implies $
> P_{\text{RS}}(\ge 1 \text{ relevant}) > P_{\text{unif}}(\ge 1 \text{ relevant}).
> $
>
> Hence, Step 2 strictly increases the probability that a random path uses at least one relevant feature, provided SHAP assigns more total mass to relevant features than their uniform share $r/d$.
>
> **Proof sketch 2 (Step 3)**
>
> Zou [2] shows that adaptive LASSO (weighted LASSO with data-dependent weights, as in our case) has the “oracle property”: it is variable-selection consistent and asymptotically equivalent to the estimator that knows the true support, provided irrelevant variables receive asymptotically larger penalties than relevant ones.
>
> In our notation, irrelevant rules receive a diverging penalty, while relevant rules receive a bounded penalty. This matches the sufficient conditions for the adaptive LASSO oracle property in [2]. Consequently, Step 3 has strictly better asymptotic selection properties than vanilla (unweighted) LASSO.
>
> ---
>
> # Question 2
>
> Valence framing, oversimplification, and information overload are widely documented as key mechanisms in misinformation spread and cognitive bias (§1). Our focus is *not* to exhaustively cover all possible biases, but to study three mechanisms that:
>
> * have clear, measurable *output-side* proxies (length, readability, sentiment/subjectivity), and
> * are plausibly driven by *global beliefs* (e.g., “controversial topics require more hedging” or “complex topics get simplified”).
>
> We deliberately designed three increasingly complex families of injected rules (§5). Our tiers are deliberately simple but already push existing global XAI methods to failure on non-convex cases (Table 5). We emphasize this more strongly in §7 as evidence that *even simpler-than-real* heuristics are challenging, so our performance should be viewed as a lower bound on difficulty.
>
> Notably, we do not know whether actual LLM bias is more likely to align with the third category (non-convex) bias.
>
> ---
>
> # Question 3
>
> We agree that the paper would benefit exemplary, visual, and qualitative material. Due to space, these were relegated to the appendix (e.g., Table 3, App. K examples).
>
> ---
>
> # References
>
> [1] Lundberg SM, Lee SI. *A unified approach to interpreting model predictions*. NeurIPS, 2017.
>
> [2] Zou H. *The adaptive lasso and its oracle properties*. Journal of the American Statistical Association, 2006.

---

### Author Response · Authors · 2025-11-29

Dear program/area chairs,

We’ve just read the emails regarding the OpenReview Data Leak issue and we are very sorry to hear of the incident. Given the situation and the fact that you’re going to re-assign Area Chairs, we’ve a question to ask.

Would it be possible to have this paper reviewed by at least one reviewer whose confidence/expertise score on the main topics of the paper (i.e., rule-based XAI) is 4 or higher, so that the technical details can be properly assessed?

At present, all assigned reviewers have indicated a confidence score lower than or equal to 3, i.e., that they are not familiar with the related work and (to quote verbatim the definition of confidence score 3) “Math/other details were not carefully checked.” In fact, some reviewers have even indicated confidence scores of 1 and 2. As a result, the technical content has effectively not been evaluated by a domain expert.

The rebuttal phase was meant to clarify these details so that the reviewers could make a more informed decision, but they can no longer respond (and have not done so thus far). Consequently, this paper has effectively not been reviewed by a domain expert, which we are not sure is consistent with the standards of an A* conference.

We understand that the large volume of submissions makes situations like this increasingly common, and that the systemic issues AI conferences are undergoing are not the fault of the reviewers or area chairs. If it is not possible to find an expert in (global) rule-based XAI as an additional reviewer for this paper, we will of course understand. However, if feasible, we would kindly request that such a reviewer be assigned so that the technical content can be fairly evaluated.

Best regards,
The Authors

---

### Meta-Review · Area_Chair_78hg · 2026-01-06

**Summary:**

Reviewers raised a variety of concerns. Some of those that were discussed in detail include lack of strong theoretical grounding for the work; weak justification for bias taxonomy; concerns around circularity of argument (particularly, use of llm-based belief scores); lack of clarity around SHAP background perturbations; as well as insufficient error analysis and failure modes.

A second cluster of concern raised by reviewers pertaining to real world relevance of the work (particularly, the extent to which the proposed method can detect real-world bias); multiple reviewers noted that the belief abstraction layer that converts textual behavior into numerical variables risks discarding semantic nuance; the reliability of faithfulness metric and to what extent high MRR guarantees human-meaningful explanations; as well as the scope of interpretability (global vs local) with reviewers pointing out that RuleSHAP remains limited to global heuristics in addition to an interpretability gap for subtle biases, for example in the case of sarcasm or metaphors all mark substantial concerns.

**Reviewer Concerns:**

Most of the first cluster of concerns (including lack of strong theoretical grounding for the work; weak justification for bias taxonomy; concerns around circularity of argument (particularly, use of llm-based belief scores); lack of clarity around SHAP background perturbations; as well as insufficient error analysis and failure modes) were addressed satisfactorily.

Where as the second cluster of concerns (including the real world relevance of the work (particularly, the extent to which the proposed method can detect real-world bias); multiple reviewers noted that the belief abstraction layer that converts textual behavior into numerical variables risks discarding semantic nuance; the reliability of faithfulness metric and to what extent high MRR guarantees human-meaningful explanations; as well as the scope of interpretability (global vs local) with reviewers pointing out that RuleSHAP remains limited to global heuristics in addition to an interpretability gap for subtle biases, for example in the case of sarcasm or metaphors all mark substantial concerns) remain largely outstanding.

**Reviewer Scores:**

The paper received 3 x Rating 4 (marginally below the acceptance threshold) and a rating of 6 (marginally above the acceptance threshold) from the other reviewer.

---

### Decision · Program_Chairs · 2026-01-26

Reject